# TGF-β in Hepatic Stellate Cell Activation and Liver Fibrogenesis—Updated 2019

**DOI:** 10.3390/cells8111419

**Published:** 2019-11-11

**Authors:** Bedair Dewidar, Christoph Meyer, Steven Dooley, Nadja Meindl-Beinker

**Affiliations:** 1Department of Medicine II, Medical Faculty Mannheim, Heidelberg University, 68167 Mannheim, Germany; bedair.dewedar@pharm.tanta.edu.eg (B.D.); christoph.meyer@medma.uni-heidelberg.de (C.M.); steven.dooley@medma.uni-heidelberg.de (S.D.); 2Department of Pharmacology and Toxicology, Faculty of Pharmacy, Tanta University, 31527 Tanta, Egypt

**Keywords:** TGF-β, liver fibrosis, hepatic stellate cells, myofibroblasts

## Abstract

Liver fibrosis is an advanced liver disease condition, which could progress to cirrhosis and hepatocellular carcinoma. To date, there is no direct approved antifibrotic therapy, and current treatment is mainly the removal of the causative factor. Transforming growth factor (TGF)-β is a master profibrogenic cytokine and a promising target to treat fibrosis. However, TGF-β has broad biological functions and its inhibition induces non-desirable side effects, which override therapeutic benefits. Therefore, understanding the pleiotropic effects of TGF-β and its upstream and downstream regulatory mechanisms will help to design better TGF-β based therapeutics. Here, we summarize recent discoveries and milestones on the TGF-β signaling pathway related to liver fibrosis and hepatic stellate cell (HSC) activation, emphasizing research of the last five years. This comprises impact of TGF-β on liver fibrogenesis related biological processes, such as senescence, metabolism, reactive oxygen species generation, epigenetics, circadian rhythm, epithelial mesenchymal transition, and endothelial-mesenchymal transition. We also describe the influence of the microenvironment on the response of HSC to TGF-β. Finally, we discuss new approaches to target the TGF-β pathway, name current clinical trials, and explain promises and drawbacks that deserve to be adequately addressed.

## 1. Introduction

### 1.1. Liver Fibrosis and Hepatic Stellate Cell (HSC) Activation

Liver fibrosis is the common result of chronic liver damage from any etiology, such as alcoholic steatohepatitis (ASH), non-alcoholic fatty liver disease (NAFLD), hepatitis B or C infection (HBV/HCV) or cholestatic liver injury, although the scar pattern formation may vary in dependency of the location of the primary damaged hepatocytes. Fibrosis means that an excessive amount of fibrillar extracellular matrix (ECM) proteins, e.g., collagen I and III, is deposited in the space of Disse [1,2]. The changes in ECM composition induce liver sinusoidal endothelial cells (LSEC) to loose fenestra and to form a basement membrane in a process termed LSEC capillarization [3]. These pathophysiological changes interfere with normal nutrient transport between sinusoidal blood and surrounding cells, especially hepatocytes, which leads to functional disturbance [1,3]. If the liver damage-inducing agent is not properly removed, liver fibrogenesis perpetuates (reflected by the term “chronic”) until the liver architecture is strongly distorted, and can then progress to the late stages of liver disease, liver cirrhosis and hepatocellular carcinoma (HCC), and thus could finally lead to liver failure and death [4,5,6].

Myofibroblasts (MFB) are the main producers of collagens and other ECM proteins and are therefore central in scar formation during liver fibrogenesis. The origin of MFB is intensely debated. MFB may originate from bone marrow-derived fibrocytes, portal fibroblasts, or hepatic stellate cells (HSC), depending on the type of damage and location. It has been proposed that liver epithelial cells, i.e., hepatocytes and cholangiocytes, or even endothelial cells could be additional sources for liver MFB, through epithelial mesenchymal transition (EMT) or endothelial mesenchymal transition (EndMT) processes [7,8]. There are supportive and disproving data available, however, for the latter transdifferentiation pathways, there is a broad consensus that HSC are the major contributors for the MFB pool during liver fibrosis, independent of the damaging source [4].

In normal liver, HSC exist in a quiescent non-proliferative state, where they have a star-like shape with intracellular lipid droplet storage containing vitamin A as retinyl palmitate [9]. During acute and chronic liver injury, transforming growth factor (TGF)-β is activated from deposits in the ECM and expressed and released from various cell types. An important target of TGF-β in this setting are HSC that are induced to activate and transdifferentiate to MFB, which comprises loss of intracellular vitamin A droplets, adaptation of a fibroblast shape, and development of a contractile, proliferative, and migratory phenotype (Figure 1) [9].

In cooperation with other signaling pathways, triggered by e.g., reactive oxygen species (ROS), platelet-derived growth factor (PDGF), and connective tissue growth factor (CTGF), TGF-β signaling is considered the key fibrogenic pathway that drives HSC activation and induces ECM production [8,10]. In normal liver, quiescent HSC express a minute amount of TGF-β, which is upregulated shortly after liver injury. Besides HSC, there are additional cellular sources for TGF-β in the liver such as LSECs, macrophages, and hepatocytes [11]. Very recently, platelets were also described as an important source of TGF-β during liver fibrosis [12].

### 1.2. TGF-β Family

The TGF-β family comprises 33 members including TGF-βs, activins, and bone morphogenetic proteins (BMPs) [13]. TGF-β proteins exist in three isoforms, TGF-β1, 2, and 3, which share overlapping but non-redundant functions, as most importantly shown by knockout (*KO*) mice of the respective TGF-β isoforms [14,15,16]. Generally, TGF-β1 is the most widely and most intensely investigated isoform in liver fibrogenesis [10,17]. Therefore, when using the term TGF-β, the knowledge is based on experimental data from TGF-β1 throughout this review. In this regard, we previously reported a predominant role for TGF-β2 in the pathogenesis of biliary fibrosis. In liver tissue of bile-duct ligated (BDL) and *Mdr*2^-/-^ mice, TGF-β2 was elevated to a higher level than TGF-β1 [18]. Moreover, a TGF-β2 antisense oligonucleotide efficiently attenuated biliary fibrosis, as shown by reduced sirius red and α-SMA staining (data under revision). Abd El-Mequid et al. describe a correlation between TGF-β2 expression in peripheral leucocytes and protein levels in serum with hepatic fibrogenesis in a cohort of 89 HCV patients and 21 healthy controls. The authors thus suggest TGF-β2 as a promising biomarker for liver fibrosis in HCV [19]. Mechanistically, the increased expression of TGF-β2 in HCV-induced liver fibrosis is mediated via a cAMP-responsive element-binding protein (CREBH) site in the promoter of the TGF-β2 gene in hepatocytes that is activated from HCV infection [20].

### 1.3. TGF-β Signaling

TGF-β is synthesized in the form of a latent precursor that needs to be cleaved by furin-like proteases to become activated. Subsequently, the C-terminus of the TGF-β molecule binds to the N-terminus of the latency associated protein (LAP) to form the latent TGF-β complex (LTC) [21]. The LTC is then released and deposited in the surrounding ECM through binding to latent TGF-β binding protein (LTBP), forming the large latent complex (LLC) [22]. All TGF-β isoforms undergo such process, whereas certain BMPs and activins are not released as latent complexes [13]. Entrapment of TGF-β in the form of LLC ensures focused local effects upon activation. The release of active TGF-β from LLC is actively triggered through variant physical and biochemical factors, such as high or low pH, cleavage by specific proteases, and through interaction with integrins. The integrin interactions are considered the principal activating mechanism for latent TGF-β. Through integrin binding to certain collagen molecules in the ECM, traction forces are applied, which induce a conformational change in the LLC, thereby facilitating the accessibility of proteases that mediate the liberation of the active form of TGF-β [22,23,24,25]. We recently identified extracellular matrix protein 1 (ECM1) as a stabilizer of liver ECM deposited latent TGF-β by interacting with αv integrins. In different animal models of experimental liver fibrosis and in patients with chronic liver diseases (CLD), ECM1 expression decreased along with disease severity. *Ecm1-KO* mice spontaneously developed severe liver fibrosis with tremendous TGF-β/Smad3 and subsequent HSC activation. The animals die between 8 and 12 weeks of age. This phenotype could be rescued by adenoassociated virus (AAV) mediated expression of ECM1 or by interfering with TGF-β signaling using AAV expressing soluble TβRII. Moreover, carbon tetrachloride (CCl_4_)-induced liver damage was blunted by ECM1 overexpression [25]. 

Active TGF-β starts signaling by binding to the TGF-β type II receptor (TβRII) resulting in recruitment of the TGF-β type I receptor (TβRI). Next, TβRII phosphorylates TβRI at a Gly-Ser–rich (GS) domain leading to a conformational modulation in TβRI and sensitizing it to bind and phosphorylate its substrates, i.e., SMAD2 and SMAD3 proteins (also called receptor-activated SMADs or R-SMADs). After C-terminal SMAD phosphorylation, pSMAD2 and pSMAD3 form heterocomplexes with the common SMAD4, which thereafter translocates to the nucleus to bind DNA and regulate the transcription of multiple target genes, e.g., *αSMA*, and *CTGF* (Figure 2) [13,26]. Two important facts deserve to be highlighted here. First, SMAD2 does not bind to DNA, while SMAD3 possesses a weak DNA binding affinity. Therefore, SMAD2/3/4 complexes generally recruit additional transcriptional coactivators to stabilize transactivation complexes [13,27]. Second, several TGF-β target genes can be activated by R-SMADs without the requirement of SMAD4 [28].

Canonical R-SMAD-mediated TGF-β signaling does not explain all observed effects of TGF-β. Many studies identified other signaling pathways that could be activated by TGF-β, such as mitogen-activated protein kinase (MAPK), mammalian target of rapamycin (mTOR), phosphatidylinositol-3-kinase/AKT, and Rho GTPase pathways (Figure 2). TGF-β non-canonical pathways provide a broad window for intracellular cross-talk [29,30,31] and can be classified into three major groups [29]: (I) R-SMADs interact with other pathways instead of directly transmitting the signal to the nucleus. Such interaction is illustrated by the ability of SMAD2 and SMAD3 to activate ERK and PKA [32,33]. (II) TheTβR complex can activate intracellular substrates other than SMADs, such as Daxx, a proapoptotic adaptor protein, leading to JNK activation and apoptosis [34]. (III) R-SMADs could be activated by TβR-independent mechanisms. The latter mechanism is best exemplified by phosphorylation of the linker domain of R-SMADs, e.g., by ERK, which interferes with R-SMAD nuclear translocation [35]. Non-canonical pathways provide one explanation for the versatile effects of TGF-β signaling and its dichotomal functions, as for example described in carcinogenesis [36]. In fibrosis, however, such events have not yet been thoroughly investigated, with exception of linker phosphorylation [37]. It should be emphasized here that results obtained from SMAD4 *KO* cells or specific kinase inhibitor treatments should be carefully attributed to non-SMAD signaling for several reasons [29,30]. Firstly, as previously mentioned, SMAD4 is not required for transcription of several specific R-SMAD dependent genes such as *SNAI2*, *TPD52*, and *CDKN1A* [28]. Secondly, chemical inhibitors can block several kinases dose-dependently [30]. Therefore, in our opinion, specific SMAD2 and SMAD3 *KO* models represent the best way to characterize non-SMAD pathways downstream to TGF-β treatment [29]. Signaling kinetics can also be utilized to shed light on SMAD and non-SMAD-dependent effects. For example, in some cells, e.g., mast cells, TGF-β mediated ERK phosphorylation occurs within 10 min in similar kinetics to EGF-induced ERK activation, which suggests a direct non-SMAD mechanism. In contrast, the same effect requires several hours in other cells, e.g., pancreatic acinar cells, highlighting the need for de novo protein synthesis, which could rely on SMAD dependent mechanisms [30,38,39,40]. Additional examples of non-canonical TGF-β signaling in the context of liver fibrogenesis are mentioned in different sections of this review.

#### 1.3.1. SMAD2 vs. SMAD3

Although SMAD2 and SMAD3 share 92% of their amino acid sequence, they show distinct biological effects. As already mentioned above, SMAD2, in contrast to SMAD3, has no DNA binding capacity, as it has extra sequences inserted in its Mad homology (MH1) domain, which prevents DNA binding [27]. This difference in DNA binding ability provides one explanation for the variant functions of the two SMAD molecules in general, in the different liver cell types and during liver fibrosis [27]. SMAD3 is considered the crucial inducer of the fibrogenic program in HSC, whereas SMAD2 was described as having antifibrotic effects [41,42]. A mechanism mediating such antifibrotic effects of SMAD2 was recently deciphered by Xu et al. and was linked to its ability to increase ligand (TRAIL) mediated HSC apoptosis through downregulation of X-linked inhibitor of apoptosis protein (XIAP). This then leads to enhanced caspase-3 activity and cell death [42]. In addition, deletion of SMAD3, but not SMAD2, in fibroblasts efficiently reduced pressure overload-induced cardiac fibrosis [43], which again underlines the differential roles of SMAD2 and SMAD3 in mediating organ fibrosis.

#### 1.3.2. SMAD Phosphorylation Dynamics

Noteworthy, the peak of a transient TGF-β induced SMAD phosphorylation generally occurs within 30–60 min, turning back to baseline at between 3 and 6 h. On the other hand, the full differentiation of MFB needs between 2–3 days [26,44]. An extended pSMAD kinetics investigation by Ard et al. shows that SMAD2 remains partially phosphorylated for up to 2 days after TGF-β stimulation in human lung fibroblasts. The authors claim that this sustained SMAD2 phosphorylation, besides transcription of SMAD3-dependent genes, is required for full differentiation of MFB [45]. These results suggest a co-operative rather than an antagonistic relationship between SMAD2 and SMAD3 in the context of MFB trans-differentiation, at least in the lung. Data from liver in this respect are lacking but given the cellular context-dependency of TGF-β signaling, a similar mechanism can be expected. Extended kinetic studies in HSC thus may generate valuable data. 

## 2. Regulation of the TGF-β Pathway

Cells keep TGF-β signaling under tight control by a plethora of positive and negative feedback loops. One of these regulatory mechanisms is the transcription of inhibitory SMAD proteins (I-SMAD) i.e., SMAD6 and SMAD7 [46], which can be induced by TGF-β as a negative feedback system. Of note, SMAD6 is selective for BMP receptors, whereas SMAD7 can bind to and inhibit both TGF-β and BMP receptors [27,47]. I-SMADs negatively regulate and fine-tune TGF-β signaling through several mechanisms, including competitive inhibition of R-SMAD binding to TβRI, recruiting ubiquitin E3 ligases, e.g., SMURF2, to TβR and induce its degradation, or phosphatase enzymes, e.g., PP1C, for receptor deactivation [27,48,49]. RING finger protein 11 (RNF11) positively regulates TGF-β signaling by competing with SMAD7 for binding to SMURF2, thus decreasing degradation activity for receptor complexes [50]. On the other hand, the deubiquitination of SMAD7 by OTU domain-containing protein (OTUD) 1 stabilizes SMAD7, thus leading to enhanced negative regulation of TGF-β signaling [51]. Moreover, the sumoylation of SMURF2 by SUMO E3 ligase PIAS3 enhances its ability to degrade TGF-β receptors and thereby suppresses TGF-β effects, e.g., EMT [52].

To get a broader overview of the currently known regulatory mechanisms of TGF-β signaling, the reader is referred to the elegant review by Derynck and Budi [13]. In the next section, we will discuss more recently identified or more deeply characterized targets and effectors of the TGF-β pathway that were described in the context of HSC activation and liver fibrosis.

### New Targets and Regulators of the TGF-β Pathway in Liver Fibrosis

Caveolin is an essential component of the caveolae of the plasma membrane and is involved in cell surface receptor trafficking [53]. Receptor-mediated endocytosis is another basic mechanism for the internalization of various proteins e.g., TGF-β receptors from cell surface through formation of clathrin-coated vesicles [54]. These two endocytosis processes were described to regulate TGF-β signaling differentially. In contrast to clathrin-dependent endocytosis of TGF-β receptors, which facilitates TGF-β signaling, caveolin mediated endocytosis promotes TGF-β receptor degradation [55] and relates to non-Smad TGF-β signaling [56]. *Cav1* deficiency was found to aggravate CCl_4_-induced liver fibrosis in mice, which was mechanistically linked to enhanced TGF-β induced oxidative stress [57]. Further, a Cav1 scaffolding domain (CSD) peptide attenuated CCl_4_-induced liver fibrosis as shown by significantly decreased collagen content [58]. The authors showed that *Cav1* knockdown led to enhanced TGF-β signaling and SMAD2 phosphorylation. In addition, SMAD2 phosphorylation decreased after CSD treatment. Further, our group identified a crucial role for Cav1 towards hepatocyte apoptosis. Cav1 levels determine whether hepatocyte apoptosis is executed upon a TGF-β stimulus [59]. Based on the regulation of Cav1 expression in hepatocytes in vitro [60] and in vivo upon NAFLD development [61], Cav1 serves as a crucial regulator of TGF-β signaling and thus TGF-β-related phenotypes.

TGF-β-HAS2-HA: Seki and coworkers delineated another downstream route of TGF-β mediated HSC activation in liver fibrosis, which ends in hyaluronan (HA) production, a major ECM glycosaminoglycan, and biomarker of liver cirrhosis. They show that HA synthase 2 (HAS2) is transcriptionally up-regulated by TGF-β and mediates HSC activation by Wilms tumor 1, CD44, Toll-like receptor 4 (TLR4), and Notch1. HAS2 expression is elevated in human and murine liver fibrosis. Furthermore, HA production and liver fibrosis were decreased upon HAS2 depletion in HSC and enhanced upon HAS2 overexpression. Therapeutical blunting of HA synthesis by 4-methylumbelliferone decreases HSC activation and liver fibrosis [62], which may offer a new therapeutic route in future.

TGF-β-CD147 feedback loop: CD147 is a glycosylated protein that is expressed on the cellular membrane of HSC [63]. Li et al. described an interesting positive feedback loop between TGF-β1 signaling and CD147 in HSC. On one side, TGF-β1 increased expression of CD147 via a SMAD2/3/4-dependent mechanism, which can promote LX-2 cell migration and contraction. On the other side, CD147 overexpression triggered TGF-β1, α-SMA, and COL1α1 expression through upregulation of ERK1/2 and Sp1 [64]. The study raises the possibility that a combination of TβR inhibitors and an anti-CD147 antibody could be more effective than TβR inhibitors alone in treating liver fibrosis. However, more studies on this topic are warranted to determine effectiveness.

Hydrogen peroxide-inducible clone 5 (HIC-5) was previously identified as a gene induced in response to increased H_2_O_2_ and TGF-β1 and was shown to be essential for MFB differentiation [65,66]. Noteworthy, the expression of HIC-5 protein increased in liver fibrosis in mice and humans [67]. HIC5 knockdown attenuated liver fibrosis in CCl_4_ and BDL animal models through upregulation of SMAD7 expression [67]. Yet, the molecular mechanisms regulating HIC-5 expression during liver fibrogenesis and their link to TGF-β signaling (components, e.g., SMAD7) are still waiting further investigation.

GP73 is a type II transmembrane glycoprotein, which is highly expressed in hepatocytes during acute and CLD [68]. GP73 was recently described as a novel target and regulator of the TGF-β pathway. Its promoter contains a SMAD binding element (SBE) which is responsible for enhanced transcription upon stimulation with TGF-β [69]. Subsequently, GP73 feeds back on decision making between canonical and non-canonical arms of TGF-β signaling, thereby downregulating SMAD signaling and facilitating ERK/AKT signaling [69]. The modulatory effect of GP73 on TGF-β signaling was attributed to GP73 mediated upregulation of caveolin that subsequently modulates TGF-β signaling as discussed above [69]. However, the in vivo relevance of GP73 in HSC activation and liver fibrosis awaits further studies.

Galectin-1 (Gal-1) is a β-galactoside-binding lectin, which recognizes and binds to several glycosylated receptors leading to the regulation of diverse intracellular signaling pathways [70]. Gal-1 expression was increased in ethanol-fed HCV core-transgenic mice and participated in MFB activation in cancer [71,72]. The link to TGF-β signaling and HSC activation was shown by a recent study by Wu et al. [73]. Treatment of LX-2 cells with recombinant Gal-1 protein increased phosphorylation of SMAD2, SMAD3, and ERK1/2 leading to enhanced HSC migration. These pro-migratory properties of Gal-1 were found to be dependent on Gal-1 binding to neuropilin-1 in a glycosylation-dependent manner [73]. Based on these results, it will be interesting to test specific Gal-1 binding inhibitors in animal models of liver fibrosis.

Lipin-1 is a phosphatidic acid phosphatase enzyme that catalyzes phosphatidate conversion to diacylglycerol [74]. It also acts as a transcriptional factor by direct binding to other transcriptional factors including PPAR-γ coactivator-1 α (PGC-1α) and PPAR-α [75]. Lipin-1 was reported to have antifibrogenic effects, which are mediated by inhibition of TGF-β1 signaling [76]. Jang et al. showed that TGF-β keeps lipin-1 downregulated on protein level in activated HSC through induction of lipin-1 polyubiquitination without altering lipin-1 mRNA stability [76]. Interestingly, resveratrol, a polyphenolic compound with antioxidant properties inhibited TGF-β-mediated polyubiquitination of lipin-1 which led to a suppression of TGF-β induced expression of fibrogenic genes [76]. Whether this is a unique response to resveratrol or represents a common mechanism for other antioxidants remains an interesting open question.

Notch and Y-box binding protein (YB-1): Recent studies describe new roles for Notch signaling and YB- signaling in regulating TGF-β signaling. Wu et al. show that both TGF-β and Notch signaling were activated in concanavalin A-induced liver fibrosis in rats [77]. Inhibition of Notch signaling using γ-secretase inhibitor reduced TGF-β and SMAD3 expression at protein and mRNA levels in peripheral blood mononuclear cells of fibrotic rats [77]. In our laboratory, treatment of primary and JS-1 HSC with Delta-like (DLL)-4 notch ligand did not alter TGF-β-induced HSC activation [78]. However, we showed that DLL4 ligand could modulate CCl_4_- and BDL-induced liver fibrosis through inhibition of inflammatory CCL-2 chemokine secretion from macrophages [78]. These results suggest that Notch signaling alters liver fibrogenesis in TGF-β dependent and independent ways. Recently, Xiong et al. described an interesting mutual interaction between YB-1 protein and TGF-β signaling in liver fibrosis. On one hand, the TGF-β signaling pathway could induce YB-1 protein expression via a SMAD2-dependent mechanism. On the other hand, phosphorylated YB-1 acts as a stabilizer for SMAD2 protein through inhibition of ubiquitination [79].

**NUAK1** and **2** are members of the AMP-activated protein kinase (AMPK) family. Both are upregulated in response to TGF-β by a SMAD2/3/4-MAPK-dependent mechanism [80]. Kolliopoulos and colleagues described a NUAK1/2 bifurcation loop in the TGF-β pathway, with NUAK1 suppressing and NUAK2 enhancing TGF-β signaling [80]. The authors show that NUAK2 is physically associated with SMAD3/TβRI, leading to stabilization of the activated receptor complex, which results in enhanced TGF-β signaling. Opposite to NUAK1 overexpression results, siRNA mediated silencing of NUAK2 attenuated TGFβ mediated α-SMA protein expression and TGF-β induced collagen gel contraction in AG1523 cells, suggesting a potential fibrogenic role [80].

**STAT3/JAK1** We and others previously revealed a cross-talk between TGF-β and STAT3 in HSC [81,82,83,84]. A recent study describes a novel mechanism by which TGF-β activates STAT3 downstream signaling involving JAK1. The authors show that JAK1 associates with TβRI and mediates TGF-β induced activation of STAT3 by two distinct mechanisms. First, an early and direct SMAD independent phosphorylation of STAT3 occurs. Second, late phosphorylation of STAT3 requires SMAD activation and *de novo* protein synthesis. The authors further show that rapid SMAD independent activation of STAT3 is essential for expression of TGF-β targets genes, such as *PDGFB*, *ID1*, *SMAD7*, and *SOCS3* in LX-2 cells, thus priming the cells for a second enhanced activation. Therewith, they suggest a co-operative interaction of both mechanisms in mediating TGF-β effects in liver fibrosis [82]. In contrast, Wang et al. showed that activated STAT3 physically interacts with SMAD3 leading to a disruption of SMAD2/3/4 complex formation and thus to attenuation of TGF-β signaling in cancer cells [83]. From these findings, it is concluded that the outcome of the interaction between TGF-β signaling and STAT3 is strongly cell-context dependent.

Tetratricopeptide repeat domain 3 (TTC3) is a ubiquitin E3 ligase, which is transcriptionally induced by TGF-β by a SMAD2/3 dependent mechanism. TTC3 then positively regulates TGF-β signaling through ubiquitination and by triggering proteasomal degradation of other ubiquitin E3 ligases, i.e., SMURF2, a well-known negative regulator of TGF-β signaling that mediates ubiquitination and subsequent proteasomal degradation of SMAD2/3 and TβRs [85]. In line, Kim et al. showed that TTC3 siRNA suppressed TGF-β induced MFB differentiation and EMT, whereas TTC3 overexpression, in the absence of TGFβ, induced EMT and MFB differentiation [85].

Mesothelin (MSLN) is a membrane-anchored cell surface protein and described as novel marker for activated portal fibroblasts [86,87]. The deletion of MSLN resulted in upregulation of *Thy-1* expression and formation of an inhibitory complex between THY-1 and TβRI, resulting in attenuation of TGF-β induced portal fibroblast activation. Even more, blocking antibodies against MSLN were able to suppress BDL mediated cholestatic fibrosis [88]. 

BMPs belong to the TGF-β superfamily and signal through SMAD1/5/8 complexes [13,89]. BMPs have diverse effects on liver fibrosis development and progression. For example, we recently reported upregulation of BMP9 expression in HSC and determined its fibrogenic role in the chronic CCl_4_ mouse liver fibrosis model as well as in human patients with CLD, whereas BMP2 was proven to have antifibrotic effects in CCl_4_ and BDL models of liver fibrosis [90,91]. Further, BMP-2 suppressed expression of TGF-β, TβRI and TβRII in HSC, whereas exogenous TGF-β and TGF-β signaling decreased BMP-2 expression, suggesting the existence of mutual regulation between the pathways of these two cytokines [91]. In addition, gremlin1, a TGF-β target gene, acts as antagonist of BMP7. It was upregulated in the porcine serum-induced hepatic fibrosis model [92]. Taking into consideration the antagonistic functions of BMP7 and TGF-β in liver fibrosis, gremlin1 could act as a bridge connecting these two signaling pathways [10].

ArfGAP with GTPase domain, Ankyrin repeat and PH domain (AGAP) 2 is a GTPase activating protein, which was recently described as a target of TGF-β signaling in human HSC. AGAP2 knockdown interfered with recycling of TβRII to the cell membrane and thus diminished TGF-β signaling and downstream effects in LX-2 cells, including proliferation, migration, and profibrogenic gene expression, e.g., *ACTA2, COL1A2, LOX, PDGFB*, and *TGFΒ2* [93].

Factor VII-activating Protease (FSAP) is synthesized and released mainly from hepatocytes and acts as a regulator of fibrinolysis and coagulation [94,95]. FSAP expression is decreased in mouse and human fibrosis [96]. In addition, single nucleotide polymorphisms in *HABP2*, the gene encoding FSAP, resulted in decreased FSAP enzymatic activity and is associated with the severity of liver fibrosis in patients, probably due to decreased FSAP-mediated degradation of PDGF-βa potent HSC mitogen [95,97]. Recently, a link to the TGF-β pathway was uncovered by Leiting et al. demonstrating a dose- and time-dependent TGF-β-mediated inhibition of *Habp2* mRNA and protein expression via a SMAD2-based mechanism [95].

Wnt: Beljaars et al. and Spanjer et al. reported a cross-talk between Wnt and TGF-β signaling [98,99]. TGF-β, but not other cytokines like PDGF, IL-1, or TNF-α, enhanced the expression of *WNT5A*, a major Wnt ligand in LX-2 cells [99]. Furthermore, TGF-β induced the expression of the Wnt receptors *FZD2* and *FZD8* in these cells [98,99]. These results indicate that the TGF-β pathway utilizes Wnt signaling to achieve HSC activation, but also acts in a paracrine way. Supporting this notion, *WNT5A* RNAi decreased TGF-β induced expression of *COL1A1*, *COL3A1*, *FN*, and *VIM* in HSC [99].

## 3. TGF-β Activity and the Microenvironment in Liver Fibrosis

First, we want to define what we understand as microenvironment in the liver and more specifically, upon liver damage. Let us assume the hepatocytes represent the center of the liver that sends out signals that vary in stages of physiology/homeostasis or damage. These signals are then translated from the microenvironment into distinct actions (matrix, inflammatory, etc.), which then feedback to the respective hepatocytes. In this way, the microenvironment is composed of the non-parenchymal cells (NPC), which are initially HSC, KCs, and LSEC, but also later infiltrated populations of inflammatory cells, the respective composition of the ECM, the pressure of the bloodstream and the biliary system. A thus defined microenvironment is highly dynamic upon hepatocellular damage along the different progression stages of hepatitis, fibrosis, cirrhosis, and cancer. Cellular fate changes driven by secreted factors and de novo activated intracellular signaling pathways induce the respective tissue responses, like cell proliferation, cell migration, tissue inflammation, and ECM deposition respective scar formation. Hereby, TGF-β is a critical cytokine triggering canonical and non-canonical intracellular pathways leading to activated HSC, macrophages with variant polarization and LSEC capillarization. These processes can be described as major features of the microenvironment, driven by and in exchange with stressed hepatocytes (reviewed in [2,3,100,101,102,103,104,105,106]).

### 3.1. Composition of the ECM

During liver fibrosis, the composition of the hepatic ECM changes, among others from collagen type IV and laminin towards collagen types I and III [9,107]. This change is sensed by HSC through two types of collagen receptors, i.e., discoidin domain-containing receptors and the integrins [108,109]. Two recent studies investigated the consequences of integrin deletion from fibroblasts and hepatocytes on TGF-β signaling. In the study by Bansal et al., integrin α11 *KO* fibroblasts show impaired proliferation, contractility and reduced responsiveness to TGF-β [110]. In the second study, integrin αvβ8 depletion from hepatocytes protected the cells against the antiproliferative effects of TGF-β, leading to enhanced liver regeneration after partial hepatectomy [111]. The mechanisms behind these interesting observations yet need to be clarified.

Kindlin-2 is an adapter protein that mediates cell-matrix and cell-cell adhesion and has been identified as a crucial regulator of integrin activation [112,113]. It is upregulated in the livers of patients with CLD and in the mouse model of chronic CCl_4_-related liver damage. In LX-2 cells, TGF-β increases kindlin-2 expression via non-canonical p38 and MAPK-dependent signaling. On the other hand, Kindlin-2 overexpression enhances TGF-β induced phosphorylation of SMAD2/3, suggesting a mutual regulation between TGF-β and Kindlin-2 [113].

Another modulated ECM component in fibrogenesis, which is related to TGF-β signaling and HSC activation, is fibronectin. Fibronectin exists in an insoluble cellular form as well as in a soluble form that is released into circulation by hepatocytes [114,115]. Cellular fibronectin has two splice variants, ED-A and ED-B [116,117]. During fibrogenesis, TGF-β and ED-A-fibronectin establish a positive feed-forward cycle. That is, TGF-β modulates ED-A splicing of fibronectin and promotes its expression. Further, ED-A *KO* fibroblasts display a reduced response to active TGF-β [118]. It was also reported that blocking fibronectin deposition through pUR4, a fibronectin assembly inhibitor, improved liver function and decreased collagen accumulation in mouse models of liver fibrosis [119,120]. Mechanistically, fibronectin can enhance TGF-β signaling (i) by binding to integrins, thus generating a pulling force to release encaged TGF-β, and (ii) by sequestering LTBP-1 in the ECM [115,117,121]. In line with this, blocking antibodies against the ED-A domain impaired incorporation of LTBP-1 into the ECM and consequently decreased TGF-β activation [115]. On the other hand, we found, rather contradictory, increased active TGF-β concentrations and enhanced HSC activation and proliferation in hepatocyte-specific *Fn-KO* mice [119,120]. These conflicting results suggest a more complex interplay between TGF-β and fibronection that needs further investigation.

An additional profibrogenic micro-environmental factor is the collagen triple helix repeat containing 1 (**CTHRC1**), which is secreted from HSC during liver fibrosis. Although CTHRC1 can activate Wnt and TGF-β pathways in parallel, the CTHRC1-mediated activation of quiescent HSC mainly depends on TGF-β signaling. Recombinant CTHRC1 activates canonical TGF-β/SMAD2/3 signaling in rat HSC. Treating HSC with a TβRII neutralizing antibody or a TβRI inhibitor suppressed CTHRC1 mediated induction of α-SMA expression, whereas a WNT3A neutralizing antibody had no effect [122]. Antagonizing CTHRC1 function thus may be a promising avenue to interfere with HSC activity in chronic liver diseases.

### 3.2. Matrix Stiffness

Matrix stiffness is not considered as an endpoint of organ fibrosis anymore, but rather as a critical regulator of the process itself. In line, there are reports showing that matrix stiffness precedes fibrosis [123,124]. HSC change their phenotype in response to stiffness and softness of the surrounding ECM [125]. Transfer of activated HSC from stiff plastic dishes to collagen-coated dishes decreased the expression of *TGFB1*, *α-SMA*, and *ED-A* [126]. Matrix stiffness also modulates TGF-β signaling, mainly via three mechanisms: (i) activation of ECM-entrapped latent TGF-β, (ii) stimulation of TGF-β non-canonical signaling pathways, and (iii) mechano-activation of HSC. For further discussion of these mechanisms, the reader is referred to the recent review by Santos and Lagares [123].

The degree of matrix stiffness is controlled by two groups of cross-linking enzymes. One group comprises the lysyl oxidase (LOX) family that is responsible for cross-linking elastin and collagen by oxidation [127]. β-aminopropionitrile (BAPN), an irreversible LOX inhibitor, inhibited HSC activation and amplified fibrosis reversion in a CCl_4_ mouse model of liver fibrosis [128]. Stimulation of LX-2 cells with TGF-β increased mRNA and protein expression of LOX-like (LOXL)-1 in a SMAD2/3-dependent mechanism. Furthermore, *LOXL1* siRNA suppressed TGF-β effects on proliferation and activation markers in HSC [129]. Noteworthy, LOX is discussed as a serum biomarker of liver fibrosis in patients with NAFLD [130].

Another enzyme that essentially contributes to matrix cross-linking is transglutaminase (TG)2. TG2 regulates TGF-β signaling by cross-linking latent TGF-β binding protein (LTBP) to the matrix, and thus mediates entrapment of latent TGF-β in the ECM [131,132]. Its role in CLD, however, needs to be determined.

### 3.3. TGF-β and Inflammatory Cells

Inflammatory cell infiltration, HSC activation, and their subsequent removal are critical features of the healing process in response to an acute liver injury. During chronic liver injury, clearance of the liver from activated HSC and inflammatory cells, in particular macrophages, is dysregulated due to the constitutive occurrence of the liver cell-damaging insult [133]. Although the reasons underlying such dysregulation are still not fully understood, several mechanisms have been delineated and are obvious. Lodyga et al. very recently reported new insights. In their work, they show that cadherin-11 (CDH11) provides intercellular junctions between activated HSC and macrophages. As a consequence, TGF-β is produced and secreted from macrophages and activated in close proximity of connected HSC, which thus show prolonged activation. The authors suggest formation of a profibrotic niche and spatial targeting for active TGF-β by this mechanism [134]. Inhibition of the CDH11 could disrupt this niche and enhance the fibrosis resolution process.

With focus on inflammatory cytokines, Cai and coworkers identified CXCL6 to be upregulated in serum and liver tissue of liver fibrosis patients. Treatment of the HSC line HSC-T6 with CXCL6 did not mediate their activation directly. Instead, CXCL6 induced TGF-β production in KC. The authors further show that HSC are activated from the supernatant of CXCL6 treated KC via a SMAD3-BRD4 complex that binds to the *c-Myc* promoter. Subsequent binding of c-MYC to the *Ezh2* promoter then leads to profibrogenic gene expression. Parts of these findings were confirmed in a CCl_4_-CLD mouse model [135].

Fabre et al. reported increased IL-17 and IL-22 production in advanced liver fibrosis, whereby in human patients, neutrophils and mast cells were the main sources of IL-17. IL-22 was identified as a profibrogenic cytokine by enhancing non-SMAD TGF-β/p38 MAPK signaling in HSC. The data were then confirmed with IL-22 RA1-*KO* mice that display less fibrosis in both thioacetamide (TAA) and CCl_4_ models of CLD. Also, inhibiting IL-22 or IL-17 production by aryl hydrocarbon receptor or RAR-related orphan receptor γ antagonists improved fibrosis [136].

Natural killer (NK) cells mediate liver fibrosis reversion by inducing interferon (IFN)-γ mediated death of activated HSC. Shi et al. examined the frequency, phenotype, and function of NK subsets in HBV-driven liver disease. NK subsets were reduced in number and displayed less antifibrotic activity in cirrhotic patients as compared to earlier disease stages. Based on in vitro studies, the authors characterized TGF-β as important factor to (1) decrease antifibrotic activity of NK cells and (2) to deplete hepatic NK cells by an HSC mediated emperipolesis (a cell-in-cell structure) and subsequent apoptosis. This process was inhibited by anti-TGF-β treatment. These data were confirmed by colocalization studies of α-SMA+ cells and NK cells in patients. The authors, therefore, suggest an anti-fibrotic approach by enhancing NK cells activity [137].

### 3.4. TGF-β and Pathophysiological Blood Flow in Liver Fibrosis

Besides matrix stiffness, fluid pressure of blood and bile in the liver vessels represents an additional mechanical parameter that influences adjacent cells with modulation of biochemical signals that translate into cellular fate decisions. Swartz and colleagues recently showed that a slow fluid flow stimulates HSC activation through the upregulation of TGF-β [138]. Physiological fluid flow upregulated profibrotic genes, e.g., *TGFB*, *ACTA2*, *Col1A1* in oral and dermal fibroblasts, in the absence of any exogenous mediators [138,139]. Interestingly, a physiological fluid flow augmented cytoplasmic internalization of caveolin and nuclear translocation of the TβRII in dermal fibroblasts [139]. The latter effects could partly explain why exogenous TGF-β combined with flow state did not lead to further augmentation of dermal fibroblast activation and even antagonized the effects of the physiological fluid flow in this study. The relevance of flow in CLD has yet to be determined, even more considering the sinusoidal structure and the impact of LSEC (defenestration). 

### 3.5. Dynamics of TGF-β Ligand Availability

The strength of fibroblast responsiveness is determined, among others, by the amount of TGF-β, which is released from cells upon *de novo* synthesis in response to a distinct stimulus and the amount which is already stored in the surrounding milieu and activated. Ansorge et al. showed that fibroblasts get activated to a similar extent by a lower concentration of a “sustained release form” of TGF-β (achieved by binding of TGF-β to agarose microbeads), as compared to treatment with active recombinant cytokine in traditional cell culture experiments [140]. It is suggested that this type of TGF-β release in constitutive portions resembles more the in vivo situation than the traditional cell culture treatment approach. Using mathematical modeling based on experimental data, it was as well predicted that continuous or pulsating TGF-β treatment will induce different cellular responses [141]. It was also shown that increasing medium volume while keeping a constant number of TGF-β molecules per cell delayed TGF-β ligand depletion and reduced a switch-like response of the TGF-β pathway, as described mathematically by Hill’s coefficient [141,142].

As mentioned above, TGF-β is produced as an inactive latent complex, enveloped by its pro-peptide LAP, and then deposited in the ECM. For receptor binding and signaling, TGF-β must be released from the complex and the matrix, e.g., to initiate ECM synthesis in HSC. One factor able to induce latent TGF-β activation by cleavage is plasma kallikrein. Subsequently, the N-terminal cleavage product of LAP can be detected in vicinity of activated HSC [143]. The authors now developed an ELISA-based method to also identify the C-terminal LAP cleavage product (LAP-DP) and showed that it can be used as a blood-based biomarker for fibrogenesis. In line, LAP-DP levels were increased in both CCl_4_ and BDL fibrosis models in mouse, prior to an increase in hydroxyproline. Furthermore, LAP-DP levels correlated with α-SMA expression [144].

## 4. TGF-β Signaling, Cell Damage and Oxidative Stress in Liver Fibrosis

ROS plays crucial roles in liver fibrosis and HSC activation [9,145]. The two main enzyme families generating ROS during liver injury are nicotinamide adenine dinucleotide phosphate (NADPH) oxidases (NOX) and the CYP450 family members [8]. ROS can act as both inducer or effector of the TGF-β signaling pathway, thereby generating a vicious cycle for fibrosis [146,147]. On the one hand, ROS induces TGF-β signaling through different mechanisms such as (i) activation of matrix metalloproteinases, (ii) induction of TGF-β expression, and (iii) augmentation of TGF-β release through activation of LAP [8,146]. On the other hand, TGF-β augments mitochondrial ROS production by activating the mTOR pathway and reducing the activity of complex III and IV [147]. Interestingly, mTOR kinase inhibitor rapamycin was recently reported to increase ROS production as well as to activate latent TGF-β which led to upregulation of CTGF in hepatic progenitor cells through SMAD2, but not SMAD3 [148]. The relationship between NOX1, 2, and 4, HSC activation, and TGF-β signaling were described previously [8,149]. NOX1 and NOX2 were increased in CCl_4_ and BDL mouse models of liver fibrosis. Their knockdown attenuated hepatic fibrosis and ROS generation [150]. A recent report also showed upregulation of NOX1 in non-alcoholic fatty liver disease (NAFLD), which impaired the hepatic microcirculation through formation of protein nitrotyrosine adducts and by reducing NO availability [151]. NOX4 was also upregulated in mouse and human liver fibrosis and was essential for HSC activation and maintenance of MFB fate [152]. Recently, involvement of NOX5 in HSC activation was demonstrated as well [153]. Overexpression of *NOX5β* and *NOX5ε* splice variants upregulates *COL1A1* and elicits the proliferation of human HSC (LX-2), whereas *NOX5* knockdown inhibits these effects [153]. Interestingly, TGF-β stimulated *NOX5* expression in a ROS-dependent mechanism [153]. The role of TGF-β/NOX5 signaling in human fibrosis remains to be elucidated.

A new antagonistic player in the liver fibrosis scene is nuclear factor-erythroid 2-related factor 2 (Nrf2). Nrf2 signaling represents a cellular protection mechanism that is induced in cells exposed to oxidative stress. Nrf2 regulates the transcription of various antioxidant enzymes such as glutathione S-transferase, glutathione reductase, glutathione peroxidase, and heme oxygenase-1 [154]. Moreover, the Nrf2 pathway to some extent protects the liver against toxin-induced fibrosis [155]. An association between TGF-β, ROS, and Nrf2 was reported for the fibrogenic processes of several organs, including the liver [154]. Sulforaphane, an Nrf2 activator, inhibited TGF-β signaling and reduced hepatic fibrosis in the BDL model [156]. *Nrf2*-deficient HSC were much stronger activated by TGF-β treatment as compared to wild type cells [157]. Another Nrf2 activator, tBHQ, was also shown to attenuate intestinal fibrosis by inhibiting the ROS-dependent TGF-β/SMAD pathway [158].

Recently, Schisandrin B, the main bioactive ingredient of the Chinese herb “Schisandra chinensis”, was reported to improve liver fibrosis in the CCl_4_ mouse model by regulating Nrf2 and TGF-β/SMAD signaling pathways [159]. Of particular interest, ascorbic acid, an antioxidant with Nrf2 inhibitory activities, promoted TGF-β mediated HSC activation via a SMAD2/3-independent mechanism [160,161]. The full mechanism of the profibrogenic effect of ascorbic acid is still unclear but there are hints for a relation to epigenetics [160,162].

In alcoholic liver disease, alcoholic hepatitis (AH) represents a life-threatening disease complication with significantly reduced liver function. Identification of molecular mechanisms that lead and drive (acute) alcoholic hepatitis are lacking since there are no available animal models to phenocopy the disease setting. Aergemi et al. analyzed livers from patients with variant stages of AH by RNA sequencing. One major finding was the lack of transcription factors involved in the determination of physiological hepatocyte fate. Most interestingly, they identified TGF-β as major upstream driver of metabolic and anabolic dysfunctions. Mechanistically, there is evidence that profound changes in DNA methylation and chromatin modification are involved in severe AH [163].

## 5. TGF-β Signaling and Epigenetics in Liver Fibrosis

Epigenetics stands for qualitative and quantitative reversible changes in gene expression without affecting the DNA sequence itself [164]. Epigenetic regulatory mechanisms include microRNAs (miRNAs), long non-coding RNA (lncRNAs), circular RNA (circRNAs), RNA binding protein (RBP), as well as DNA and histone modifications. During recent years, much evidence has been gathered showing epigenetic regulation of TGF-β signaling. On the one hand, TGF-β induces epigenetic changes in HSC e.g., by regulating expression levels of distinct pro- and antifibrotic miRNAs [165,166]. On the other hand, epigenetic regulators modulate the activity and expression of up- and downstream components of the TGF-β signaling pathway. In the following section, we will discuss the recent findings interconnecting TGF-β signaling with epigenetics.

miRNAs are short non-coding nucleotide sequences (19–23 nucleotides), which regulate gene expression at the posttranscriptional level by binding to 3’-UTRs of target mRNAs. Upon mRNA binding, miRNAs induce either their degradation or interfere with their translation [167]. miRNA profiles of human HSC change upon culture activation [168] and many results reported the modulatory roles of miRNAs on the expression of liver fibrosis genes. Thus, several miRNAs were classified as antifibrotic and related to the TGF-β pathway, such as miR-19b, miR-34a-5p, miR-146a, miR-133, and miR-134. For example, miR-133 is downregulated in HSC during murine and human liver fibrogenesis and upon TGF-β treatment. Overexpression of miR-133 in HSC, in contrast, inhibits collagen expression [169]. miR-193 and miR-30c are TGF-β dependently downregulated in cultured HSC and in experimental liver fibrosis models, and potentially target TGF-β2 and SNAIL1 [170]. Other miRNAs have been described as profibrogenic, e.g., miR-942 and miR-125b [171,172,173,174,175,176,177]. In the past, different mechanisms were delineated how miRNAs may regulate TGF-β signaling during fibrogenesis. All these miRNAs mediate their effects by targeting various mediators of the TGF-β pathway. miRNA-30, miRNA-212-3p, and miRNA-17-5p modulate TGF-β signaling strength by targeting the *SMAD7*. While *SMAD7* is a direct target of pro-fibrotic miRNAs 17-5p [178] and 212-3p [179], miRNA-30 inhibits expression of SMAD7 inhibitor *KLF11* [180]. miRNA-9-5p targets TβRI and TβRII mRNA in LX-2 cells and is itself downregulated by TGF-β through promoter methylation [165]. TβRII expression is also downregulated by augmenter of liver regeneration (ALR) under healthy conditions in LX-2 cells, whereas TGF-β treatment induces miRNA-181a, which silences ALR expression [181]. Further, it was demonstrated that miR-19b and miR-142-3 target TβRI. As another TGF-β signaling component, SMAD4 is targeted by miR-34a-5p and miR-146a [171,172,173,182]. Microinjection of miRNA-455-39 suppressed heat shock protein factor 1 (*Hsf1*) in livers of CCl_4_, BDL and high-fat diet (HFD) fed mice, therewith improving fibrosis. Mechanistically, miRNA-455-3p binds to the 3’UTR of *Hsf1*. Downregulated *Hsf1* reduces expression of heat shock protein 47, thus inhibiting TGF-β signaling [183]. miRNA-134 has also decreased in the CCl_4_ and BDL models or after HFD feeding in rats. This miRNA exerts its function in human and rat HSC by binding to the 3´ untranslated region of TGF-β activated kinase 1-binding protein 1 (*TAB1*), therewith interfering with its TGF-β dependent induction, which finally results in decreased HSC proliferation, α-SMA and collagen expression [175]. In portal fibroblasts, MLK1 silencing reduces TGF-β dependent expression of profibrogenic genes. Fan et al. concluded from their results that MLK1-SMAD3 interaction at the DNA levels is required for induction of TGF-β target genes. In line, MLK1 deficiency ameliorates BDL-induced fibrosis [184]. Also, SMAD3 mediated the upregulation of miRNA-31 in fibrotic samples from human and rat livers as well as in activated HSC [185]. In a very recent study, we could demonstrate that profibrogenic effects of miR-942 are mediated through targeting BAMBI, a TGF-β decoy receptor, leading to enhanced fibrogenic TGF-β signaling [176]. Further, Genz et al. showed an antifibrotic role for miR-25-3p, mediated by its ability to inhibit TGF-β induced SMAD2 phosphorylation and collagen deposition in LX-2 cells [186].

LncRNAs: In contrast to miRNAs, lncRNAs comprise longer non-coding nucleotide sequences (more than 200 nucleotides). LncRNAs regulate gene expression by acting as miRNA sponges or by competing with miRNAs for binding to mRNA, thus blocking their effects on respective mRNA targets [187,188]. Cellular availability of lncRNAs is also dysregulated through HSC activation and liver fibrosis. Long intervening noncoding RNA-p21 (lincRNA-p21) is upregulated in CCl_4_-induced liver fibrosis and acts as endogenous competitive inhibitor of miR-30, which inhibits TGF-β signaling through targeting *KLF11* and by increasing SMAD7 expression. Therewith, lincRNA-p21 enhances TGF-β signaling [189]. LncRNA H19 is also upregulated in CCl_4_ and BDL induced liver fibrosis [190,191]. Mechanistically, lncRNA H19 stabilizes TβRI by abrogating the inhibitory effect of miR-148a on ubiquitin-specific protease 4 (USP4) [192].

CircRNAs are a new class of lncRNAs with a similar mechanism of action [193]. The circRNA signature is dysregulated upon radiation-mediated HSC activation [194]. Zhou et al. reported a significantly increased expression of several circRNAs, e.g., mmu_circ_33594, mmu_circ_34116, and mmu_circ_35216 in fibrotic livers and in a mouse HSC cell line (JS1) upon TGF-β stimulation [195]. The mechanistic link between TGF-β and these circRNAs still needs further investigations.

RBPs bind to specific RNA motifs, which results in modulation of their transcription, localization, and stability. RBPs, miRNAs, lncRNAs, and circRNAs together form an RNA regulon that controls the expression of numerous genes [196]. Recently, insulin-like growth factor 2 binding protein 3 (IGF2BP3) was identified as an RBP and to be an effector and target of TGF-β signaling. *Igf2bp3* knockdown decreased the effects of TGF-β on HSC activation, whereas TGF-β signaling inhibition induced *Igf2bp3* expression in a miRNA dependent mechanism, therewith revealing an interesting positive feedback loop between TGF-β and IGF2BP3 [197].

DNA and histone modification: DNA methylation and histone acetylation represent additional abundant epigenetic mechanisms that are currently intensely investigated [198,199] as presented in the following.

DNA methylation: Transcription of multiple genes including fibrosis-related genes can be repressed by methylation of cytosine-phosphoguanine (CpG) dinucleotides in their promoters through DNA methyltransferases (DNMT) [199]. One of these genes is angiogenic factor with G patch and FHA domains 1 (*Aggf1*), which is downregulated in different models of liver fibrosis and in activated HSC through increased CpG methylation of DNA surrounding the *Aggf1* promoter. 5-Azacytidine, a DNMT inhibitor, restored *Aggf1* expression, which led to improvement of liver fibrosis in a SMAD7-dependent mechanism [200]. In line, 3-deazaneplanocin, another DNMT inhibitor, improved CCl_4_ induced liver fibrosis in mice [201]. Recently, enhancer of zeste homolog 2 (EZH2), which methylates the lysine residues 9 and 27 of histone 3 was found to play a crucial role in experimental liver fibrosis through modulation of TGF-β signaling. Indeed, TGF-β specifically increased the expression of EZH2 without influencing other histone-lysine N-methyltransferase enzymes. In addition, inhibition of EZH2 diminished TGF-β-induced transcription of profibrogenic genes e.g., *Fn, Col1a1*, and *Acta2* [202]. Noteworthy, various EZH2 inhibitors e.g., MAK683, SHR2554, and CPI-1205 are currently investigated in clinical trials for their potential in the treatment of different oncologic diseases (ClinicalTrials.gov Identifier: NCT02900651, NCT03603951, NCT03741712, and NCT03480646). Our group showed synergism between SMAD2/3 and TRIM33, which is essential in directing liver progenitor cell differentiation towards hepatocytes [203]. TRIM33 is a chromatin reader that prepares the cell genome for transcriptional activity by modulating DNA methylation, thereby relevant in determining cellular fate. Interestingly, TRIM33 owns super affinity for phosphorylated SMAD2/3 in competition with SMAD4. We found that TRIM33 formed a transcription factor complex in combination with *p*-SMAD2/3 in liver progenitor cells. The complexes are essential for expression of key hepatic functional genes, e.g., albumin and coagulation factors, and largely determine whether LPCs are capable of taking over hepatic function in acute-on-chronic liver failure patients suffering from massive parenchymal necrosis [203].

Histone acetylation and deacetylation: During HSC activation and liver fibrosis, histone deacetylation catalyzed by histone deacetylase (HDAC) is also crucial in regulating the expression of various profibrogenic genes [198]. Also, in this case, HDAC inhibitors display antifibrotic effects in animal models of CLD. As one example, Wang and colleagues recently reported that an HDAC inhibitor, SAHA, improved liver function and decreased hepatic fibrosis in rats [204]. SAHA mediated its antifibrotic effects by increasing SMAD7 expression, which leads to an attenuation of TGF-β signaling by enhanced negative feedback [204]. Furthermore, P300 acetyltransferase is long known to promote TGF-β signaling through acetylation of histones and SMAD2/3 [205,206,207]. Very recently, P300 was additionally shown to enhance TGF-β signaling by a non-canonical mechanism via acting as a shuttle to facilitate the nuclear transport of the SMAD2/3 complex [208]. In this study, the authors showed that p300-mediated histone acetylation enhanced the response to TGF-β in HSC [208]. In line, Jiang and his colleagues have shown that nicotinamide riboside, which increases the intracellular pool of NAD+, decreased the expression of P300, whereas it increased the activity of Sirt1, an NAD+-dependent histone deacetylate enzyme, leading to inhibition of SMAD2/3 acetylation and attenuation of TGF-β signaling [209]. Interestingly, thyroid hormone triiodothyronine (T3) inhibited TGF-β1-mediated histone 4 acetylation at the promoters of TGF-β1 target genes, i.e., *Smad7, Id1*, and *p15* leading to suppression of TGF-β signaling [210]. The antifibrotic effects of T3 were demonstrated in liver and skin fibrosis models, which were induced by CCl_4_ and bleomycin, respectively [210].

## 6. TGF-β and Mesenchymal Transition in Liver Fibrosis

EMT is a reversible process during which epithelial cells lose polarity and change their phenotype to a mesenchymal fate [7]. Three basic types of EMT have been described so far. EMT type 1 and type 3 are connected to embryogenesis and cancer metastasis, respectively, whereas EMT type 2 is associated with wound healing, organ fibrosis and tissue regeneration [7]. Although epithelial liver cells, i.e., hepatocytes, and cholangiocytes, are able to undergo EMT in vitro, the existence of EMT in vivo is highly debated [211]. In a recent review article by Weng et al., it was hypothesized that EMT-like features in patients with advanced CLD could be a side effect of the TGF-β enriched microenvironment [211,212,213,214,215].

Noteworthy, EMT is a dynamic process characterized by transitional states. After a complete EMT, cells have totally lost their epithelial character and have acquired a mesenchymal phenotype. However, some cells only undergo partial EMT, thus presenting both epithelial and mesenchymal markers [7]. Recently, Wu et al. reported that TGF-β or activin signaling induces a partial EMT in hepatic progenitor cell (HPC) derived cell lines via the canonical SMAD signaling pathway [216]. One can conclude that cellular fate changes of HPC due to partial EMT could facilitate their contribution to liver fibro-carcinogenesis. In cancer cells, sustained TGF-β exposure induces a stable form of EMT through non-Smad mTOR signaling [217]. A sustained SMAD2 phosphorylation for more than 6 h, for example, was reported to be necessary for a complete trans-differentiation of human fibroblasts [45]. In contrast to a reversible EMT, which is crucial for wound healing processes and tumor dissemination, a more stable EMT phenotype is only partially reversible and contributes to tumor persistence and latency [217]. This study raised an interesting hypothesis on the existence of a robust EMT as a feature of liver fibro-carcinogenesis, especially pinpointing on the fibrotic tumor microenvironment that presents with high amounts of TGF-β for a prolonged time. 

Sun and coworkers claim to have identified a mechanism for the occurrence of hepatocyte EMT during liver fibrosis. They treated human L02 cells and primary rat hepatocytes with advanced oxidation protein products (AOPP) and subsequently tested for EMT features. They found reduced E-cadherin and increased vimentin expression, collagen deposition and induced cell migration. AOPP also increased cellular ROS levels and activated TβRs and Smad signaling. Interfering with ROS production and TGF-β signaling blunted AOPP mediated hepatocyte EMT [218].

Mesothelial cells (MCs) form a single layer of the mesothelium and cover the liver surface. It was suggested that in biliary fibrosis upon bile duct ligation (BDL) or in CCl_4_-induced fibrosis in mice, MCs may migrate inside the liver and contribute to the population of activated HSC/MFB by a process termed mesothelial mesenchymal transition (MMT) [219]. Li and colleagues reported that this cell fate modulation requires TGF-β since this conversion was suppressed in TβRII deficient MCs in culture and in vivo. The authors further showed that such MCs derived HSC are predominantly located near the liver surface and contribute to capsular fibrosis. Based on FACS cell sorting, the authors claim that the MC-derived HSC population stores little vitamin A-containing lipid droplets presents with a fibrogenic fate and contributes to about 1.4 and 2.0% of activated HSC in the BDL and CCl_4_ models, respectively. During fibrosis reversion in the CCl_4_ model, 20% of MC-derived MFB survive and deactivate (senescent state) to vitamin A-poor HSC [220].

Interestingly, TGF-β could also induce a mesenchymal transition of endothelial cells (EC) in a process termed EndMT [221]. Ribera et al. recently reported that a small fraction of EC undergoes EndMT in the liver of CCl_4_-treated mice and they claim that this is also occurring in cirrhotic patients [222]. Inhibition of EndMT by BMP-7 improved liver fibrosis in mice [222]. In this regard, Randi and colleagues reported that endothelial transcription factor (ETS)-related gene (Erg) is essential in maintaining liver homeostasis by binding to SMAD3 and hindering its nuclear translocation [223]. Abrogation of *Erg* enhanced SMAD3 dependent TGF-β signaling in EC, resulting in induction of EndMT and the promotion of liver fibrogenesis [223].

## 7. TGF-β and Metabolic Fate Changes in Liver Fibrosis

Activated HSC are roughly characterized, among others, by α-SMA expression and to synthesize and secrete or deposit ECM. The activation process requires a metabolic reprogramming of the cells. Indeed, the HSC trans-differentiation process is energy demanding. The core metabolic changes include a switch from oxidative phosphorylation to aerobic glycolysis [224]. Although glycolysis generates less adenosine triphosphate (ATP) than oxidative phosphorylation, it requires less time, which makes this process finally more efficient to produce the required amount of ATP for HSC activation [224,225,226]. TGF-β was reported to induce glycolysis, and thus can be considered a driver of metabolic reprogramming. A very interesting mechanistic study in idiopathic pulmonary fibrosis (IPF) suggests that a TGF-β-induced glycolytic pathway leads to the accumulation of lactic acid, which acidified the pH in the microenvironment, and thus activated ECM bound latent TGF-β, therewith representing a positive feedback loop to increase TGF-β action [224,227]. Besides glycolysis, glutamine metabolism was recently shown to be essential for TGF-β induced HSC trans-differentiation [228,229]. Glutaminolysis is a two-step reaction that involves the conversion of glutamine to glutamate by glutaminase (GLS) and then to α-ketoglutarate through glutamate dehydrogenase or aminotransferase enzymes [230]. In activated HSC, TGF-β induces *Gls1* through SMAD3 and p38 MAPK. On the other side, depletion of extracellular glutamine or silencing *Gls1* in the presence of glutamine prevented TGF-β induced HSC activation and decreased the expression of profibrotic markers [229].

Interestingly, a recent study by Takahashi et al. assigned new metabolic functions to the TGF-β2 isoform. In this study, exercise-induced TGF-β2 stimulated cellular uptake of fatty acids and glucose, therewith improving insulin sensitivity [231,232]. The role of TGF-β2 and exercise on HSC activation and liver fibrosis remains an open question. So far, Takahashi et al. reported a decrease in liver mass and liver fat content after administration of TGF-β2 in high-fat diet (HFD) treated mice [231]. Also of relevance, knockout of *Smad4* stimulated β-oxidation of fatty acids and suppressed lipid-induced fibrosis and inflammation in mouse livers [233]. The distinct metabolic roles of TGF-β1 and 2 again highlight the non-redundant functions of these TGF-β isoforms.

## 8. Circadian Rhythm, TGF-β Signaling, and Liver Fibrosis

The circadian rhythm regulates numerous physiological processes that require oscillation during the daily 24 h cycle e.g., heart rate, body temperature, blood pressure, and some metabolic processes such as glycolysis [224,234]. This oscillatory behavior is driven by a network of clock genes, e.g., Period 1 (*Per1*), Period 2 (*Per2*), Period 3 (*Per3*), brain and muscle aryl-hydrocarbon receptor nuclear translocator-like 1 *(Bmal1*) and Cryptochrome (*Cry1* and *Cry2*) [224,235]. In fibrosis, several clock genes were found to play critical roles. For example, depleting *Per2* in mice strongly enhanced CCl_4_ and BDL-induced liver fibrosis, highlighting its protective role to maintain liver homeostasis [236,237]. Moreover, there is mutual regulation between TGF-β signaling and clock genes. For instance, several regulators of TGF-β signaling, such as SMURF2 and SMAD7 contain *Bmal1* binding sites in their promoter, suggesting that their transcriptional regulation has a “clock component” [238]. In HT22 neurons and NIH3T3 fibroblasts, TGF-β2 inhibits a number of clock-controlled genes including *Dbp* and *Tef*, without changing *Bmal1* levels [239]. In a recent study, TGF-β increased *Bmal1* expression in normal lung fibroblasts and in an animal model of lung fibrosis [240]. *Bmal1* knockdown abrogated TGF-β induced EMT of lung epithelial cells and inhibited differentiation of normal lung fibroblasts [240]. These results at least suggest relevance of *Bmal1* in mediating TGF-β related signals in lung fibrogenesis. The existence of a similar regulation system in liver fibrosis warrants further studies. Importantly, *Bmal1* was linked to inflammation in several settings [241,242]. Inflammation, as previously described, is a vital phenomenon preceding or accompanying liver fibrosis. Also, it needs further clarification if the aforementioned connection between TGF-β and *Bmal1* is indeed directly related to circadian rhythm.

## 9. TGF-β, Autophagy, and Senescence in Liver Fibrosis

Autophagy describes a cellular survival program during which lysosomes degrade substrates intracellularly for energy production and depletion of damaged cellular components [243]. Ligand activated nuclear receptor Rev-erb, previously shown to improve liver fibrosis in the CCl_4_ mouse model, is also characterized as a novel regulator of autophagy [244,245]. It is suggested that HSC need to undergo autophagy to provide energy for the activation process. Thomes et al. compared the impact of Rev-erb activation with a synthetic ligand, SR9009, and TGF-β treatment on autophagy during HSC activation in CCl_4_ mediated mouse fibrosis. CCl_4_ challenged mice display reduced AMPK signaling, upregulated P70S6K phosphorylation as well as increased P62 and decreased protein levels of autophagy-related gene (ATG), which is suggestive for a disturbed autophagosome formation. SR9009 prevented P70S6K phosphorylation only, whereas all other parameters were not changed. In vitro, both SR9009 and TGF-β inhibited AV biogenesis, whereas opposite effects were provided for fibrogenic gene expression, P70S6K phosphorylation, and HSC proliferation, TGF-β being an enhancer. Further, autophagy activator rapamycin and inhibitor wortmannin both antagonized HSC activation, proliferation, and P70S6K phosphorylation. In addition, inhibiting P70S6K blunted TGF-β induced HSC proliferation. The authors conclude that SR9009 and TGF-β both similarly affect autophagy, but differentially regulate HSC activation and fibrogenic fate. Based on these findings, the role of autophagy on HSC activation needs further investigation before a final conclusion can be drawn [246].

Cellular senescence provides another cellular survival mechanism during which the cell cycle is irreversibly arrested, resulting in the cease of a cell.

Replication [247]: Senescent cells are characterized by expression of p16, p21, p53, and β1-galactosidase [247]. Cellular senescence protects from malignant transdifferentiation and plays crucial roles in aging and wound healing [248]. In line, cellular senescence occurs in mouse and human liver fibrosis [249,250]. In a murine model of CCl_4_ induced liver fibrosis, Krizhanovsky et al. identified senescent cells as mainly derived from HSC. Such senescent HSC were efficiently removed by natural killer (NK) cells, suggesting that HSC senescence is a mechanism to limit liver fibrogenesis [249]. Moreover, ablation of a key senescent regulator, *P53*, resulted in excessive liver fibrosis, with a delayed resolution after chronic administration of CCl_4_ in mice [249]. In contrast, in a mouse model of IPF, the secretome of senescent fibroblasts was found to be fibrogenic, and deletion of senescent cells improved fibrogenesis [251]. TGF-β has been described as a central component of senescence-associated secretory phenotypes (SASP). In liver, TGF-β contributes to cellular senescence in acute and chronic liver injury models [252]. A number of hepatocytes undergo senescence upon acute acetaminophen (APAP) induced liver damage [253]. Furthermore, infiltrating macrophages secrete TGF-β, therewith facilitating gain of a senescent fate to neighboring viable hepatocytes. TβRI inhibition with SB525334, a small-molecule inhibitor, decreases senescence and improves survival after challenging mice with a toxic dosage of APAP [253]. In the *Mdm2-KO* model of biliary senescence, recruited macrophages and MFB secrete TGF-β and induce senescence in the surrounding hepatocytes and cholangiocytes [254]. Inhibition of TGF-β signaling using galunisertib, another compound inhibitor of TβRI, interferes with senescence transmission and improves liver function [254]. In a recent study from Razdan et al., they found that TGF-β via SMAD3, NOX4 and ROS may induce telomere dysfunction, which was demonstrated by immunofluorescence quantification of DNA damage foci. TGF-β-mediated telomere dysfunction subsequently promotes trans-differentiation of human fibroblasts to MFB, instead of inducing senescence [252]. These results once more highlight the cell context-dependency of TGF-β effects and the existence of overlapping pathways for cellular senescence and MFB transdifferentiation. Considering the key role of TGF-β in HSC activation and liver fibrogenesis, the results of Razdan et al. seem to contrast the findings of Krizhanovsky et al. Moreover, the findings underline again that extrapolation of senescence-associated results from mouse to human cells should be performed very cautiously. One important fact to consider is that mice, in contrast to humans, express telomerase in a wide range of cell lineages [247]. This in turn means that Krizhanovsky et al. possibly describe a senescence program that is independent of telomere erosion [247]. Taken together, a link between TGF-β, HSC senescence, and liver fibrosis needs to be carefully (re)-evaluated.

## 10. Targeting TGF-β in Liver Fibrosis

Reversibility of liver fibrosis is evident in patients after removal of the causative agent of damage/disease [133,255], e.g., antiviral treatment of patients with HBV infection led to regression of virus-induced liver cirrhosis [255]. Despite clear evidence for reversibility, there is still no direct antifibrotic therapy for liver fibrosis available but urgently required because removal of the fibrosis-inducing factors is often hard to achieve, or at least not completely possible, especially in the increasing CLD entities of NAFLD and ASH. Further, even upon avoidance or abrogation of the fibrosis-inducing factor, fibrosis resolution requires time. Toning down scar formation or speeding up scar resolution processes are therefore aims of a direct antifibrotic therapy [256]. Due to the major role of TGF-β in liver fibrogenesis, several studies focused on its inhibition to develop effective antifibrotic therapies. Previous attempts to target TGF-β include, for example, the use of soluble TβRII, which compete with the membrane-anchored TβRII for TGF-β binding. These soluble receptors lack the intracellular signal transduction domain and thus act as molecular sinks [257]. Wang et al. utilized the His-SUMO expression system to generate high amounts of a soluble truncated TβRII for in vivo testing. In this study, N-terminally His-SUMO-linked TβRII significantly mitigated CCl_4_-induced liver fibrosis [258]. Other attempts include the use of anti-receptor/anti-ligand antibodies to disrupt receptor-ligand interactions or the use of small molecule kinase inhibitors such as LY2157299 (also known as galunisertib) and LY2109761 to inhibit TβRI and interrupt intracellular downstream signaling [259]. Both LY2157299 and LY2109761 are small chemical inhibitors that block TβR activity. In comparison to LY2157299, which inhibits mainly TβRI, LY2109761 was reported as a dual inhibitor of TβRI and TβRII [260]. We and other groups have shown promising results for the use of galunisertib in preclinical animal models of liver fibrosis [261,262]. Other approaches inhibited TGF-β signaling indirectly through targeting LOXL or integrins, and therewith reducing release of active TGF-β from its ECM deposited latent form [108,129,263].

Additional efforts to develop TGF-β-based therapeutics include interference with nuclear translocation of the SMAD2/3/4 oligo-complex using an aminoacyl-tRNA synthetase interacting multifunctional protein 1 (AIMP1) peptide. Of note, the AIMP1 peptide did not influence TGF-β-induced phosphorylation of SMAD2 and SMAD3 but inhibited the nuclear translocation of SMAD3 by a still not identified mechanism [264]. Moreover, Zhang and his colleagues used magnolol to prevent the interaction between SMAD3 and SMAD4 and, thus, could attenuate TGF-β signaling and concanavalin A induced hepatic fibrosis [265]. In line with the above description on the distinct roles of SMAD3 and SMAD2 in organ fibrosis, a recent study reported the efficacy of a specific SMAD3 inhibitor (SIS3) in reducing fibrosis, apoptosis, and inflammation in a mouse model of kidney fibrosis [266], however, a therapeutic effect of SIS3 on liver fibrosis still remains poorly defined.

Many natural compounds with experimental antifibrotic activity integrate at some point with the TGF-β signaling pathway. We here will just mention two most recent findings. Ganai and Husain described Genistein, an isoflavonoid found in soy, as hepatoprotective in a rat model of liver fibrosis. Daily intragastric administration of Genistein for 12 weeks attenuated d-Galactosamine -induced fibrosis. Mechanistically, Genistein increased expression of SMAD7 in liver, which ultimately interfered with TGF-β/Smad signaling [267]. Similarly, praziquantel (PZQ), a schistosomicide used in the clinics for decades, has antifibrotic activity in schistosomiasis mice. In a recent mechanistic study, the authors showed that PZQ inhibits CCl_4_-induced liver fibrosis by upregulating SMAD7 in HSC. The link between PZQ, SMAD7 expression, and HSC activation was further confirmed in LX-2 and other fibroblast cell lines [268].

Unfortunately, TGF-β signaling based antifibrotic therapies have not yet been translated to human patients. The reason for this is the complexity of this pathway, and the highly dynamic, context and cell-type dependent outcome of TGF-β signaling. Further, TGF-β nearly acts ubiquitously in the whole organism thus making all organs target of TGF-β-directed therapies., Finally animal models of liver fibrosis do not reflect all aspects of human disease [257,269]. Depending on the model, mice may need 4–6 weeks of treatment to develop liver fibrosis, whereas, in humans, the process usually takes decades [270]. Another reason for translation failure is the lack of non-invasive biomarkers that could stratify patients with liver fibrosis and aid in monitoring the response of the patients during therapeutic intervention. In this regard, Kojima and his colleagues developed an antibody against LAP degradation products (LAP-DP) that represents a measure for active TGF-β availability in serum and tissue [143,271]. Such discoveries are highly encouraging, as they can provide real-time monitoring of safety and efficacy issues, thus making translation more reliable. The broad biological effects of TGF-β already indicate that a simple and general inhibition of the ligand or the receptor complex will induce unwanted effects that sometimes my override the desired beneficial outcome. Here, more specific targeting of disease stage-related regulatory parameters and cell type-directed approaches based on basic research findings need to be developed. Recently, Qiao et al. reported an elegant strategy to differentially modulate inflammatory and fibrogenic effects of TGF-β through manipulating interactions with key transcriptional factors. Administration of ICG-001 inhibited the interaction of β-catenin with T Cell factor (TCF), therewith facilitating its complex formation with FoxO. The interaction between β-catenin and TCF is considered central for several profibrogenic pathways such as TGF-β/SMAD and Wnt/β-catenin, whereas the β-catenin/FoxO interaction mediates TGF-β induced differentiation of anti-inflammatory T regulatory (Treg) cells. Through intervention with ICG-001, Qiao et al. were able to maintain the anti-inflammatory effects of TGF-β through upregulation of T regulatory (Treg) cells, while its profibrogenic effects were suppressed in a unilateral ureteral obstruction model of kidney fibrosis. Interestingly, ICG-001 also protects the liver from profibrogenic effects of systemically administered recombinant TGF-β [272]. ICG-001 reduced HSC activation, leading to attenuation of collagen deposition, as evidenced by reduced hydroxproline content and collagen I staining in the livers upon acute CCl_4_ injury in mice. ICG-001 also displayed anti-inflammatory effects via suppression of the chemokine CXCL12 [273]. Based on these preclinical data, Kimura and colleagues nicely demonstrated tolerability of an ICG-001 analog, PRI-724, in a small group of patients with HCV induced hepatic cirrhosis [274]. A well-controlled clinical trial including more patients is now required to adequately evaluate antifibrotic effects of these small molecules during liver fibrosis. This is a good example of the possibility of uncoupling diverse TGF-β effects in a tissue, and a similar approach can be imagined for CLD. Of course, this warrants detailed knowledge of cell type and cell fate specific components of the TGF-β pathway and their signaling from the cell surface to the nucleus that are related to the respective adverse TGF-β effects. Furthermore, they need to be druggable. 

Repurposing clinically used drugs for the treatment of liver fibrosis will probably accelerate the development of new antifibrotic therapies, as these drugs have known safety profiles, and thus will facilitate the conduct of clinical trials. For example, carvedilol and metformin, approved drugs for the treatment of hypertension and type II diabetes, respectively, showed promising results in animal models of liver fibrosis [275,276]. Mechanistically, metformin suppresses TGF-β/SMAD3 signaling, thus mediating the antifibrotic effects in the CCl_4_ model of liver fibrosis in mice [277]. Unfortunately, a clinical pilot study launched in 2014 to evaluate the role of metformin in HCV-induced liver fibrosis was withdrawn in 2018 due to insufficient funding (ClinicalTrials.gov Identifier: NCT02306070). Nonetheless, hope rises again with new clinical trials that will evaluate the potential of AVID200 in patients with diffuse cutaneous systemic sclerosis and myelofibrosis (ClinicalTrials.gov Identifier: NCT03831438 and NCT03895112). AVID200 represents a computationally designed highly potent trap for TGF-β1 and TGF-β3 [278]. The hopefully positive results of these trials would then encourage further testing of AVID200 in fibrotic diseases of the liver. 

Recruitment for another study, NCT00574613, on efficacy and safety of p144, a 14mer peptide from human TGF-β1 type III receptor (betaglycan), to treat skin fibrosis in systemic sclerosis was recently completed. P144 has been specifically designed to block the interaction between TGF-β1 and TGF-β1 type III receptor, thus blocking its biological effects. P144 has shown significant antifibrotic activity in mice receiving repeated subcutaneous injections of bleomycin, a widely accepted animal model of human scleroderma. The results of this trial are expected soon.

Finally, NCT03727802, a study with TRK-250, a nucleic acid compound that inhibits the progression of experimental pulmonary fibrosis by selectively suppressing gene expression of TGF-β1, is currently (October 2019) still recruiting. The study is a double-blind, randomized, placebo-controlled phase I study. The primary objective is to assess the safety and tolerability of single and multiple inhaled doses of TRK-250 in subjects with idiopathic pulmonary fibrosis.

## 11. Conclusions and Outlook

The field of TGF-β signaling has broadened with newly identified connections to cellular programs like metabolism, ROS, senescence, circadian rhythm, epigenetics, and EMT. Continuously, new downstream branches, regulatory mechanisms and transcriptional targets of the TGF-β pathway are delineated, some of these related to HSC activation and liver fibrogenesis. However, still many aspects are not fully understood, especially with regard to a successful clinical translation from animal studies that has not been achieved yet. A major reason for this is the pleiotropic effects of this cytokine. The future aim will be to suppress excessive profibrogenic TGF-β effects, while maintaining the desired wound healing and anti-inflammatory responses, therewith providing superior TGF-β based therapeutics with minimal side effects. To date, there are no FDA approved drugs for the treatment of liver fibrosis. However, there are several ongoing TGF-β directed clinical trials conducted on patients with fibrotic diseases of other organs than the liver that, if successful, could be transferred to liver diseases and tested in well-controlled clinical trials. Due to the continuously growing understanding of the cellular context-dependency of TGF-β signaling together with advancement of diagnosis and follow up procedures for patients with liver fibrosis, we believe that TGF-β based therapeutics have a chance to become reality in the near future.

## Figures and Tables

**Figure 1 cells-08-01419-f001:**
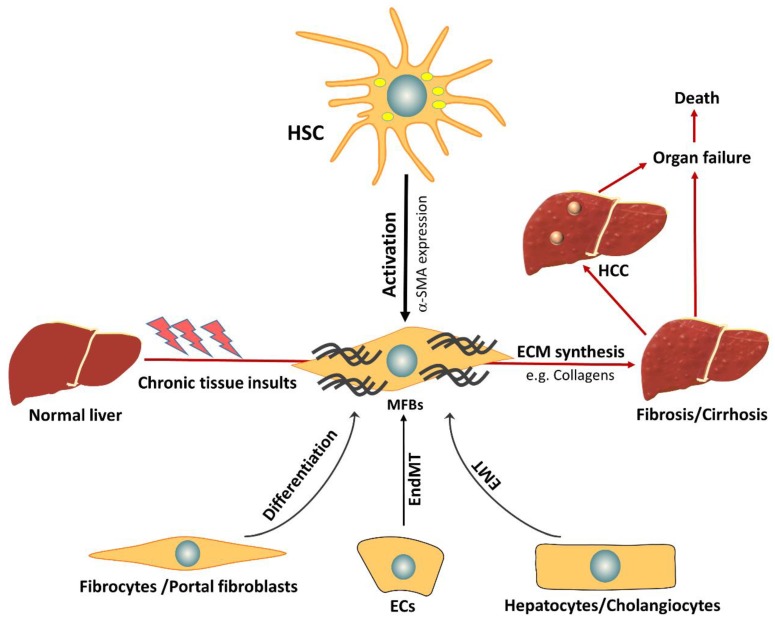
Activation of hepatic stellate cells (HSCs) and origin of myofibroblasts (MFBs) in chronic liver diseases. During activation, HSCs lose intracellular lipid droplets, acquire a fibroblast-like shape, and express a large amount of alpha-smooth muscle actin (α-SMA) and extracellular matrix proteins (ECM). Beside HSCs, which represent a major source of MFBs, other cells such as pericytes, portal fibroblasts can differentiate into MFBs. Also, endothelial cells (ECs) and epithelial cells, i.e., hepatocytes and cholangiocytes, might contribute to liver MFBs pool through an endothelial-mesenchymal transition (EndMT) and epithelial-mesenchymal transition (EMT), respectively. However, unequivocal in vivo evidence of EMT during liver fibrosis is still missing.

**Figure 2 cells-08-01419-f002:**
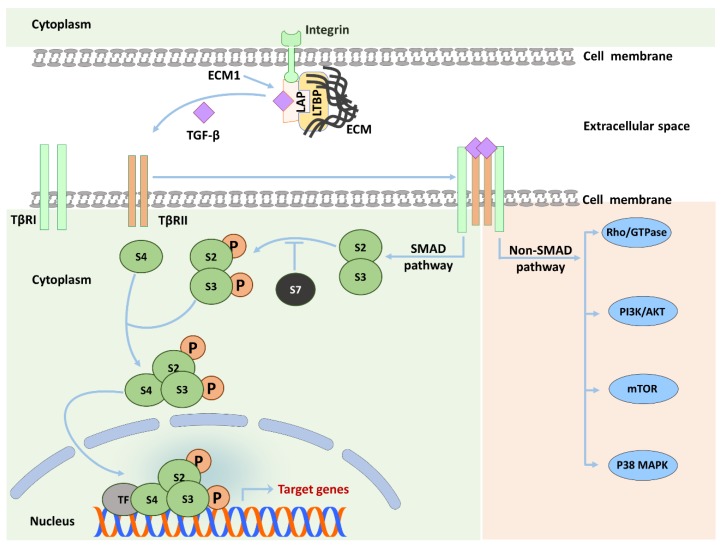
SMAD- and Non-SMAD-dependent TGF-β signaling. Upon liver damage associated signaling, TGF-β molecules are freed from the large latent complex (LLC) through the interaction of integrins with the latent association protein (LAP). Binding of released TGF-β to TβRII results in the formation of a heterotetramer with TβRI, which then initiates the canonical signaling pathway through phosphorylation of R-SMADs, i.e., SMAD2 (S2) and SMAD3 (S3). TGF-β can also activate non-canonical SMAD-independent pathways, as exemplified here by MAPK, mTOR, PI3K/AKT, and Rho/GTPase pathways. Alongside other mechanisms, SMAD7 negatively regulates TGF-β signaling through competing with R-SMADs for TβRI binding. TF: Transcription factors, P: phosphate group, LTBP: latent TGF-β binding protein.

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
