# Peer review of "TGF-β in Hepatic Stellate Cell Activation and Liver Fibrogenesis—Updated 2019"

_cells, 2019, doi:10.3390/cells8111419_

Round 1

Reviewer 1 Report

This review entitles “TGF-β in Hepatic Stellate Cell Activation and Liver Fibrogenesis – updated 2019”summarizes recent reports of TGF-β-mediated hepatic stellate cells and liver fibrosis. It seems this submitted paper is well organized with considerable effort. In terms of content, this reviewer thinks that there is almost no problem with the current version, but here are the two comments authors should consider.

In the schema of Figure 1, it is described in the flow from cirrhosis to HCC to liver failure, but there are many patients where cirrhosis leads to liver failure without going through HCC, so it appears it is better to add another line. In the “10. Targeting TGFβin liver fibrosis” (Page 21), authors described CBP-b catenin inhibitor, ICG-001 is cited for its anti-fibrotic effect in a model of nephrosclerosis. This ICG-001 has already been reported in a liver fibrosis model (Akcora BÖ et al. Biochim Biophys Acta Mol Basis Dis. 2018), and the improved PRI-724 of ICG-001 has already begun clinical trials in human cirrhosis and should be cited (Kimura et al. EBioMedicine. 2017 Sep;23:79-87) .

Author Response

please see attach

Reviewer 2 Report

The manuscript by Dewider et al is a timely review on the recent discoveries of the signalling pathways and therapeutic approaches targeting TGF-b in liver fibrosis. This manuscript is clearly written and brings valuable insight by reviewing the diverse signalling mechanisms of TGF-b in fibrosis. 

Minor comments:

Figure 2 - it is not clear whether it's meant to be two cell membranes from two different cells? if appropriate have the LLC on the same cell membrane?

The non-smad pathway is interesting- I would suggest that they authors may wish to expand on this section in the context of liver fibrosis in section 1.3

The authors may consider to first familiarise the reader with the role of TGf-b in pathophysiology (sections 3 and 4) followed by the review of the new target (section 2).

Author Response

please see attach
